# How Comorbidities Affect Hospitalization from Influenza in the Pediatric Population

**DOI:** 10.3390/ijerph19052811

**Published:** 2022-02-28

**Authors:** Sophia C. Mylonakis, Evangelia K. Mylona, Markos Kalligeros, Fadi Shehadeh, Philip A. Chan, Eleftherios Mylonakis

**Affiliations:** 1Infectious Diseases Division, Warren Alpert Medical School of Brown University, Providence, RI 02903, USA; sopmyl16@bu.edu (S.C.M.); evangelia_mylona@brown.edu (E.K.M.); markos_kalligeros@brown.edu (M.K.); fadi_shehadeh@brown.edu (F.S.); philip_chan@brown.edu (P.A.C.); 2Department of Medicine, Warren Alpert Medical School of Brown University, Providence, RI 02903, USA; 3School of Electrical and Computer Engineering, National Technical University of Athens, 15780 Athens, Greece; 4Division of Preparedness, Response, Infectious Disease, and Emergency Medical Services, Rhode Island Department of Health, Providence, RI 02908, USA

**Keywords:** influenza, children, diabetes, length of stay, metabolic diseases

## Abstract

Influenza is a contagious respiratory illness and can lead to hospitalization and even death. Understanding how comorbidities affect the severity of influenza can help clinical management. The aim of this study is to offer more information about comorbidities that might be associated with the severity of influenza in children. We used a statewide network in Rhode Island, USA, to extract data for laboratory-confirmed influenza cases among children 19 years old or younger. We identified 1169 lab-confirmed influenza cases. The most common comorbidities were asthma (17.1%), neurodevelopmental disorders (10.3%), gastrointestinal disorders (7.6%), atopic dermatitis (7%), and endocrine and metabolic diseases (6.8%). Interestingly, 80.8% (63 out of 78) of children who had an influenza-related hospital admission had at least one comorbidity, and among hospitalized children with influenza, the most common comorbidities were neurological diseases (28.2%, 22/78), gastrointestinal disorders (25.6%, 20/78), endocrine and metabolic diseases (24.4%, 19/78), and neurodevelopmental disorders (23.1%, 18/78). Children with endocrine or metabolic diseases were 8.23 times more likely to be admitted to the hospital, and children with neurological disorders were 6.35 times more likely to be admitted (OR: 8.23, 95% CI: 4.42–15.32 and OR: 6.35, 95% CI: 3.60–11.24, respectively). In summary, we identified specific comorbidities associated with influenza hospitalization and length of hospital stay, and these groups should be prioritized for public health interventions.

## 1. Introduction

In the United States, the Centers for Disease Control and Prevention (CDC) estimates that influenza affects between 9 and 45 million people (or approximately 10% of the population) annually, resulting in 140,000–810,000 hospitalizations and 12,000–61,000 deaths yearly [1,2]. Since 2010, 7000–26,000 children younger than 5 years old have been hospitalized [2,3,4,5]. Influenza can cause respiratory and non-respiratory complications, including cardiovascular complications as well as pro-inflammatory cytokines and clot formation [3].

The COVID-19 pandemic has demonstrated the impact of comorbidities on the severity of respiratory infections [3,6]. Influenza vaccination campaigns are an important component of prevention [6], and since both influenza and SARS-CoV-2 are respiratory-borne infections that cause pandemics, it is important to study the impact of comorbidities on the severity of influenza [7].

The purpose of this study was to evaluate the influence of comorbidities on the hospitalization rate of children with influenza by using the electronic network database of the largest healthcare network in Rhode Island, USA. The following comorbidities were investigated regarding their association with influenza disease severity that resulted in hospitalization and with the length of stay at the hospital: heart disease; malignancy; endocrine or metabolic diseases; asthma; respiratory, renal, and neurological disease; psychiatric, neurodevelopmental, gastrointestinal, and blood disorders; and eczema/atopic dermatitis and prematurity.

## 2. Methods

### 2.1. Data Extraction

This study was approved by the Rhode Island Hospital Institutional Review Board (IRB), and data were extracted from a statewide network in Rhode Island, USA, for the year 2018–2019. We included children younger than 19 years old, and the study was performed in line with the STROBE (Strengthening the Reporting of Observational Studies in Epidemiology) guidelines [8]. More specifically, we extracted data for demographic characteristics, such as age, gender, race, ethnicity, past medical history (ICD-10 codes), hospital admissions, and length of stay (LOS).

Our primary outcome was to identify whether demographic characteristics and specific comorbidities were associated with influenza-related hospital admissions, while our secondary outcome was to assess whether the LOS among children with influenza-related hospital admissions was associated with these variables. The main comorbidities were heart disease (including congenital malformations and ischemic and pulmonary heart diseases), malignancies (including leukemia and lymphoma), and endocrine and metabolic diseases (including diabetes, obesity, and thyroid diseases). A detailed list with diagnoses and ICD-10 codes is given in Appendix A.

### 2.2. Statistical Analysis

We compared patient characteristics using Pearson’s Chi-square test for categorical variables (median(IQR)) and Student’s *t*-test for continuous variables (*n*%). We also performed a multivariate logistic regression to examine the association between influenza-related admissions with at least one comorbidity and between influenza-related admissions and each comorbidity separately. All odds ratios (OR) were adjusted for age, gender, and race. We also performed a linear regression analysis to examine whether there was an association between the above variables and LOS. All statistical analyses were performed using Stata^®^/SE 17 (StataCorp, College Station, TX, USA). For our analyses, 95% confidence intervals (CIs) and *P* values are shown.

## 3. Results

We identified 1169 lab-confirmed influenza cases in children between 1 September 2018 and 31 August 2019. Among them, 647 (55.3%) were males. Of all confirmed cases, 516 were Non-Hispanic White, 357 were Hispanic, and 113 were Black or African American.

These cases resulted in 78 influenza-related hospital admissions. More specifically, 50 (64.1%) males were admitted, compared with 28 (35.9%) females. Overall, 80.8% of children who had an influenza-related hospital admission also had at least one pre-existing comorbidity. The most common comorbidities for children infected with influenza included asthma (17.1%), neurodevelopmental disorders (10.3%), and gastrointestinal disorders (7.6%).

Most of the admitted patients were White or Caucasian (37, 47.4%), followed by Hispanic or Latino patients (20, 25.6%) and African American patients (14, 17.9%), and the median age of the patients was 6 (IQR: 2–11) years old (Table 1). Hispanic or Latino children were 47% less likely to be hospitalized with influenza than White or Caucasian children (OR: 0.53; 95% CI: 0.30–0.95; *p* = 0.032), while Black or African American children did not show a statistically significant difference in hospital admission compared with White children. When looking at children with at least one comorbidity, Hispanic children (198/357) were 58% less likely to be hospitalized (OR: 0.42; 95% CI: 0.22–0.81; *p* = 0.009) than White or Caucasian children with comorbidities (195/516), while Black or African American children (64/113) did not show a statistically significant difference when compared with White children. More details by race group are given in Appendix A.

We performed a logistic regression and a linear regression to examine the association between influenza-related hospitalization and (1) demographic characteristics, and (2) patients that had at least one comorbidity (Table 2). Children with at least one comorbidity were 6.84 times more likely to be hospitalized with influenza (OR: 6.84; 95% CI: 3.78–12.37) compared with children without any comorbidities. The most common comorbidities were neurological disorders (28.2%), gastrointestinal disorders (25.6%), endocrine and metabolic diseases (24.4%), and neurodevelopmental disorders (23.1%).

Notably, 15 children had a diagnosis of diabetes, and 6 of them were hospitalized due to influenza; children with diabetes were 14.57 times more likely to be hospitalized with influenza compared with children without diabetes (OR: 14.57; 95% CI: 4.68–45.31). Children with other endocrine or metabolic diseases were 8.23 times more likely to be admitted to the hospital, and children with neurologic disorders were 6.35 times more likely to be admitted (OR: 8.23; 95% CI: 4.42–15.32 and OR: 6.35; 95% CI: 3.60–11.24, respectively) compared with children without any of these comorbidities. Additionally, children with heart disease; malignancies; respiratory disease; psychiatric, neurodevelopmental, blood, or gastrointestinal disorders; or renal disease were more likely to be hospitalized (Table 2).

Looking at the length of stay (LOS) in influenza-related hospitalizations (Table 3), only three diagnoses had a significant association. Specifically, the median LOS of children with heart disease was significantly longer than the LOS of children without heart disease (10 days vs. 2 days, *p* = 0.002). The median LOSs were also statistically longer among children with psychiatric disorders (6 days vs. 2 days, *p* = 0.032) and children with a premature birth (8 days vs. 2 days, *p* = 0.022).

Among children 3 years old or younger, those with endocrine or metabolic diseases were 14.76 times more likely to be admitted for influenza (OR: 14.76; 95% CI: 5.39–40.43) compared with children without endocrine or metabolic diseases. A similar association was observed for heart disease (OR: 5.19; 95% CI: 1.24–21.78), neurological disease (OR: 4.98; 95% CI: 1.78–13.98), neurodevelopmental disorders (OR: 3.33; 95% CI: 1.098–10.09), gastrointestinal disorders (OR: 6.16; 95% CI: 2.61–14.50), and prematurity (OR: 5.14; 95% CI: 1.59–16.59).

Children 4–11 years old were more likely to be admitted with influenza if they suffered from malignancies (OR: 19.12; 95% CI: 3.34–109.57), psychiatric disorders (OR: 10.91; 95% CI: 3.47–34.34), endocrine or metabolic diseases (OR: 9.66; 95% CI: 3.63–25.70), neurological disease (OR: 7.85; 95% CI: 3.28–18.79), psychiatric disorders (OR: 10.91; 95% CI: 3.47–34.34), neurodevelopmental disorders (OR: 3.37; 95%, CI: 1.40–8.14), gastrointestinal disorders (OR: 5.42; 95% CI: 1.94–15.11), or renal disease (OR: 6.91; 95% CI: 1.66–28.67). Finally, adolescents 12–19 years old were more prone to being admitted for influenza with heart disease (OR: 15.56; 95% CI: 2.69–89.81), neurological disease (OR: 8.71; 95% CI: 2.35–32.29), or blood disorders (OR: 8.63; 95% CI: 2.33–32.01) (Table 4).

## 4. Discussion

Influenza is a serious infection that leads to hospitalization and considerable morbidity among children [9]. Comorbidities have been associated with influenza severity but have not been well described [10]. We identified 1169 lab-confirmed influenza cases among children and found that comorbidities were associated with disease severity and that most hospitalized children (four out of five) with influenza had at least one comorbidity, such as endocrine or metabolic diseases, asthma, neurodevelopmental disorders, or chronic and cardiac diseases. Overall, children with at least one comorbidity were 6.84 times more likely to be hospitalized with influenza (OR: 6.84; 95% CI: 3.78–12.37). These findings suggest areas for public health intervention.

Importantly, endocrine diseases were the most frequent underlying cause of admission for influenza, but underlying endocrine, metabolic, and gastrointestinal disorders were not associated with a prolonged length of stay. Influenza affects the metabolism of human neutrophil calcium, which might be an explanation for this strong association with hospitalization. Additionally, there is mounting evidence showing that the lung is a target organ for diabetic injury from the beginning of the disease in patients of pediatric age [11,12,13,14].

In other age groups, diabetes is associated with increased mortality. For example, a study evaluating the influenza-associated diabetes mortality rate for the Mexican population found that, depending on the year, influenza-associated diabetes mortality rates for adults 20–59 years of age ranged between 1.7 and 3.4/100,000, and for adults 60 years and older, the influenza-associated diabetes mortality rates were between 16.3 and 46.1/100,000 [15]. Moreover, in a previous study that included 4306 hospitalized patients who tested positive for influenza, diabetes was associated with complicated hospitalization in the 15–50-year-old age group [16]. Our working hypothesis is that these admissions were mainly for the administration of intravenous fluids and other brief supportive care. Moreover, individuals with diabetes mellitus are more susceptible to decreased vaccine efficacy [17].

We also found that children with heart disease were more likely to be hospitalized and stay in the hospital longer due to influenza. Similarly, other studies reported increased complications in influenza infections associated with heart disease [18,19]. Although it is more common in adults, children with heart disease who are admitted to the hospital for influenza are at an increased risk for severe outcomes, such as in-hospital mortality and morbidities [20]. Unsurprisingly, children with hematologic disorders and other immune-suppression disorders were also more prone to experience severe influenza symptoms and be admitted, while children with psychiatric disorders experienced a longer length of hospital stay [21].

As noted earlier, the impact of comorbidities on the severity of viral respiratory infections has been at the forefront during the COVID-19 pandemic. Comorbidities associated with severe COVID-19 include asthma, neurological disease, obesity, and diabetes [22]. For example, among 3221 SARS-CoV-2-positive children and adolescents enrolled in a global prospective cohort study with outcome data, 3.3% had severe outcomes within 14 days [23]. Tripathi and colleagues [24] analyzed data from 795 patients (96.4% in the United States) from 45 sites, including 251 (31.5%) with obesity. Those with obesity were also more likely to be diagnosed with multisystem inflammatory syndrome in children (35.7% vs. 28.1%, *p* = 0.04) and had higher ICU admission rates (57% vs. 44%, *p* < 0.01) with more critical illness (30.3% vs. 18.3%, *p* < 0.01) [24]. Interestingly, similar to our cohort, Longmore and colleagues from the International BMI-COVID Consortium found that diabetes and obesity are independent risk factors for COVID-19 severity [25].

Importantly, our study also highlights that comorbidities may vary between cohorts. For example, influenza is associated with increased hospitalization rates among patients with asthma and chronic lung diseases [26,27]. In a cohort of 888,157 children, the adjusted incidence/1000 child-years of influenza-associated hospitalization among children with chronic lung diseases was 3.9 (95% CI: 2.6–5.2), compared with 0.7 (95% CI: 0.5–0.9) for children without chronic lung diseases [27]. Interestingly, in our study, asthma was not significantly associated with hospital admission or LOS, and this might be because it was controlled with effective management. Similarly, obesity was not a significant comorbidity in our cohort, but the strong association between hospitalization due to influenza and obesity [28] suggests that this finding should not be generalized and could be due to the retrospective nature of our study. For example, Moser et al. reported on a cohort of 4778 patients with influenza-like illness. Adults with influenza were more likely to be hospitalized if they were underweight (OR: 5.20), obese (OR: 3.18), or morbidly obese (OR: 18.40) [29]. Morgan et al. also performed a case–cohort study to evaluate whether obesity was a risk factor for hospitalization and death during the 2009 (H1N1) influenza A pandemic and found that morbid obesity may be associated with hospitalization and mortality [30].

Interestingly, Hispanic or Latino children were more than 40% less likely to be hospitalized with influenza than White or Caucasian children. This could be due to better access to testing or potential barriers to the use of health services and admission. When assessing differences in severity-adjusted pediatric hospitalization rates, Chamberlain et al. [31] found that admission rates were higher for White patients than for Non-White patients. Institutions need to monitor for health equity across race and ethnicity and remove all barriers for admission in this population.

Regarding the limitations of the study, we studied the association between comorbidities and hospitalization from infection, but our approach did not allow us to study the relative impact of each comorbidity or to study causation. Additionally, our methodology did not allow us to study elements such as resilience that may affect some diseases, or to take into account the severity of each comorbidity, which might differ from patient to patient. Furthermore, we did not account for influenza vaccination status as data were not available. Moreover, the probability of being tested could be associated with the presence of comorbidities, and influenza could be a secondary reason for hospital admission. Lastly, we used ICD-10 codes rather than questionnaires to categorize comorbidities, which may impact accuracy because of different coding criteria across individuals and institutions, and because coding criteria change over time. It should be noted that the length of hospital stay depends on different factors such as insurance coverage and the capacity for outpatient services.

## 5. Conclusions

We identified comorbidities associated with influenza-associated hospitalization and the length of hospital stay. These findings highlight the importance of influenza prevention among children in high-risk categories. Educating children through culturally-tailored interventions that are sustainable can ensure long-term decreases in the rates of hospitalization. These groups should be prioritized for public health intervention strategies on the importance of vaccination, disease prevention, and the use of face masks. Interventions and educational tools could also moderate asthma and other comorbidities such as metabolic, gastrointestinal, respiratory, and heart diseases and help prevent the need for hospitalization secondary to influenza. Further prospective studies can include the evaluation of interventions and preventive measures.

## Figures and Tables

**Table 1 ijerph-19-02811-t001:** Characteristics of children with lab-confirmed influenza.

	Total	Not Admitted	Admitted	*p*-Value
	*n* = 1169	*n* = 1091	*n* = 78	
**Age, Years**	6.00 (3.00–12.00)	6.00 (3.00–12.00)	6.00 (2.00–11.00)	0.11
**Patient Sex**				0.11
**Female**	522 (44.7%)	494 (45.3%)	28 (35.9%)	
**Male**	647 (55.3%)	597 (54.7%)	50 (64.1%)	
**Race**				**0.027**
**Black or African American**	113 (9.7%)	99 (9.1%)	14 (17.9%)	
**Hispanic or Latino**	357 (30.5%)	337 (30.9%)	20 (25.6%)	
**Other/Unknown**	183 (15.7%)	176 (16.1%)	7 (9.0%)	
**White or Caucasian**	516 (44.1%)	479 (43.9%)	37 (47.4%)	
**Comorbidities**				
**At least one comorbidity**	488 (41.7%)	425 (39.0%)	63 (80.8%)	**<0.001**
**Heart disease**	29 (2.5%)	23 (2.1%)	6 (7.7%)	**0.002**
**Malignancy**	12 (1.0%)	9 (0.8%)	3 (3.8%)	**0.011**
**Diabetes**	15 (1.3%)	9 (0.8%)	6 (7.7%)	**<0.001**
**Obesity**	13 (1.1%)	12 (1.1%)	1 (1.3%)	0.88
**Endocrine or metabolic diseases**	57 (4.9%)	38 (3.5%)	19 (24.4%)	**<0.001**
**Asthma**	200 (17.1%)	188 (17.2%)	12 (15.4%)	0.68
**Respiratory disease**	59 (5.0%)	49 (4.5%)	10 (12.8%)	**0.001**
**Neurological disease**	81 (6.9%)	59 (5.4%)	22 (28.2%)	**<0.001**
**Psychiatric disorders**	59 (5.0%)	48 (4.4%)	11 (14.1%)	**<0.001**
**Neurodevelopmental disorders**	120 (10.3%)	102 (9.3%)	18 (23.1%)	**<0.001**
**Blood disorders**	50 (4.3%)	39 (3.6%)	11 (14.1%)	**<0.001**
**Gastrointestinal disorders**	89 (7.6%)	69 (6.3%)	20 (25.6%)	**<0.001**
**Eczema/atopic dermatitis**	82 (7.0%)	75 (6.9%)	7 (9.0%)	0.48
**Prematurity**	42 (3.6%)	36 (3.3%)	6 (7.7%)	**0.044**
**Renal disease**	15 (1.3%)	12 (1.1%)	3 (3.8%)	**0.037**

Continuous data: median (IQR); categorical data: *n* (%).

**Table 2 ijerph-19-02811-t002:** Association of demographics and comorbidities with influenza-related hospital admission.

	Admission	
	OR (95% CI)	*p*-Value
**Age**	0.96 (0.92–1.00)	0.073
**Gender**		
**Female**	Reference	
**Male**	1.25 (0.76–2.04)	0.380
**Race**		
**White or Caucasian**	Reference	
**Black or African American**	1.35 (0.69–2.65)	0.388
**Hispanic or Latino**	0.53 (0.30–0.95)	**0.032**
**At least one comorbidity**	6.84 (3.78–12.37)	**<0.001**
**Comorbidities**		
**Heart disease**	3.7 (1.43–9.62)	**0.007**
**Malignancy**	4.82 (1.24–18.70)	**0.023**
**Diabetes**	14.57 (4.68–45.31)	**<0.001**
**Obesity**	1.68 (0.21–13.65)	0.626
**Endocrine or metabolic diseases**	8.23 (4.42–15.32)	**<0.001**
**Asthma**	0.81 (0.42–1.58)	0.540
**Respiratory disease**	2.80 (1.31–5.98)	**0.008**
**Neurological disease**	6.35 (3.60–11.24)	**<0.001**
**Psychiatric disorders**	4.61 (2.15–9.88)	**<0.001**
**Neurodevelopmental disorders**	2.94 (1.65–5.26)	**<0.001**
**Blood disorders**	4.4 (2.11–9.17)	**<0.001**
**Gastrointestinal disorders**	4.83 (2.72–8.58)	**<0.001**
**Eczema/atopic dermatitis**	1.25 (0.54–2.86)	0.605
**Prematurity**	2.18 (0.87–5.47)	0.095
**Renal disease**	3.77 (1.03–13.77)	**0.045**

OR: odds ratio; CI: confidence interval.

**Table 3 ijerph-19-02811-t003:** Association of various comorbidities with influenza-related LOS.

	LOS		
	Median Days (IQR)	Median Days (IQR)	*p*-Value
**Comorbidities**	**No**	**Yes**	
**Heart disease**	2 (1–4)	10 (6–30)	**0.002**
**Malignancy**	2 (1–4)	5 (1–9)	0.49
**Diabetes**	2 (1–5)	2 (1–3)	0.53
**Obesity**	2 (1–5)	3 (3–3)	0.58
**Endocrine or metabolic diseases**	2 (1–5)	2 (2–3)	0.79
**Asthma**	2 (1–4)	2 (1–5)	0.86
**Respiratory disease**	2 (1–4)	4 (2–6)	0.077
**Neurological disease**	2 (1.5–5)	2 (1–2)	0.078
**Psychiatric disorders**	2 (1–4)	6 (2–9)	**0.032**
**Neurodevelopmental disorders**	2 (1–4)	2 (2–7)	0.21
**Blood disorders**	2 (1–5)	2 (2–5)	0.78
**Gastrointestinal disorders**	2 (1–4)	2 (1–5.5)	0.9
**Eczema/atopic dermatitis**	2 (1–4)	2 (1–9)	0.96
**Prematurity**	2 (1–4)	8 (2–27)	**0.022**
**Renal disease**	2 (1–5)	1 (1–5)	0.37

**Table 4 ijerph-19-02811-t004:** Association between different variables and hospital admission for influenza by age group.

	Admission		
	OR (95% CI), *p*-Value		
	0–3	4–11	12–19
**Comorbidities**			
**Heart disease**	5.19 (1.24–21.78); **0.025**	-	15.56 (2.69–89.81); **0.002**
**Malignancy**	-	19.12 (3.34–109.57); **0.001**	-
**Diabetes**	-	-	13.85 (3.68–52.15); **<0.001**
**Obesity**	-	-	2.41 (0.26–22.42); 0.441
**Endocrine or metabolic diseases**	14.76 (5.39–40.43); **<0.001**	9.66 (3.63–25.70); **<0.001**	1.52 (0.17–13.64); 0.710
**Asthma**	2.33 (0.77–7.05); 0.134	0.46 (0.15–1.43); 0.181	0.51 (0.14–1.94); 0.326
**Respiratory disease**	2.08 (0.88–4.88); 0.094	0.58 (0.20–1.62); 0.297	0.51 (0.14–1.94); 0.326
**Neurological disease**	4.98 (1.78–13.98); **0.002**	7.85 (3.28–18.79); **<0.001**	8.71 (2.35–32.29); **0.001**
**Psychiatric disorders**	-	10.91 (3.47–34.34); **<0.001**	2.56 (0.83–7.87); 0.100
**Neurodevelopmental disorders**	3.33 (1.098–10.09); **0.034**	3.37 (1.40–8.14); **0.007**	2.10 (0.67–6.53); 0.202
**Blood disorders**	3.15 (0.79–12.60); 0.104	3.04 (0.80–11.49); 0.101	8.63 (2.33–32.01); **0.001**
**Gastrointestinal disorders**	6.16 (2.61–14.50); **<0.001**	5.42 (1.94–15.11); **0.001**	2.54 (0.62–10.37); 0.195
**Eczema/atopic dermatitis**	0.47 (0.58–3.73); 0.473	2.36 (0.81–6.85); 0.114	0.65 (0.08–5.47); 0.693
**Prematurity**	5.14 (1.59–16.59); **0.006**	0.80 (0.10–6.36); 0.832	-
**Renal disease**	-	6.91 (1.66–28.67); **0.008**	-

## Data Availability

The data presented in this study are available on request from the corresponding author. The data are not publicly available due to HIPAA and privacy regulations.

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
