# Peer review of "How Comorbidities Affect Hospitalization from Influenza in the Pediatric Population"

_ijerph, 2022, doi:10.3390/ijerph19052811_

Round 1

Reviewer 1 Report

Comments and Suggestions for Authors

Thank you for choosing me as a reviewer of this manuscript. In my opinion, the topic of this manuscript is interesting. The aim of this study is to offer more information about comorbidities that might affect the severity of influenza in children. I recommend the manuscript for publication, but after some minor revisions:

Abstract

The abstract in lines 14-15 appears, "The aim of this study is to offer more information about comorbidities that might affect the severity of influenza in children."

In lines 38-40  ,, The aim of this study was to utilize the electronic database of the largest healthcare 38 network in Rhode Island, in order to offer more information on the comorbidities that affected the hospitalization rate of children with influenza ”. I think that in the introduction the purpose should be reformulated, maybe something like "The purpose of this study is to assess(evaluate) the influence of comorbidities… using the electronic network database…"

Also, I think you need to check again the following:

- In methods, line 49, "We included children younger than 19 years old" and in the abstract row 16 ,,…. confirmed influenza cases among children 18 years old or younger ”

- lines 124-125 ,, Finally, adolescents 12-18 years old, were more prone to be admitted with influenza with heart disease…”

- table 4 row 3 "12-19"

Also, I think you need to clearly state the age group.

In lines 52-53 ,,….we extracted demographic characteristics, such as age, gender, race, ethnicity, past medical history “. Did you also discuss the resilience environment in your study? I think this can also influence the evolution of some diseases.

In lines 135-136 it appears:  ,, and found that most of the hospitalized children (4 out of 5) with influenza had at least one comorbidity…” How did you find this?

Lines 195-196: ,, Regarding limitations of the study, it is difficult to distinguish the exact impact of  each comorbidity and only a correlation between comorbidities and hospitalization from infection can be established.” Are you referring to multivariate logistic regression?

In line 204-205 : "These findings highlight the importance of influenza prevention through vaccination among children in these high-risk categories."

Did you mention somewhere in the text about the importance of vaccination against influenza in children? Did you mention any references in this regard?

Lines 205-206: “Educating children with culturally- tailored interventions that are sustainable can ensure long-term decrease in the rates of hospitalization.” What do you mean? What kind of education do children have?

Author Response

See file

Reviewer 2 Report

Manuscript ID: ijerph-1586858

 How Comorbidities Affect Hospitalization from Influenza in the Pediatric Population

Thank you for the opportunity to review this interesting study. In this manuscript, Mylonaki  et. al addressed the question how comorbidities affect hospitalization and length of stay in children with influenza. The authors extracted data from the electronic database of a large healthcare network in Rhode Island. The paper is precise and clear and the analysis appropriate. The findings are interesting and shed light in an area with limited information in the literature for the study population.

Minor comment

Lines 152-3; please rephrase the sentence

Author Response

See file

Reviewer 3 Report

In this well-written manuscript, Mylonakis S. et al. investigate the role of different comorbidities in influenza hospitalization in a relatively large cohort of children. They found that children with specific comorbidities, like endocrine diseases, belong to high-risk groups for hospitalization and long-term hospitalization due to influenza. This finding could enrich the knowledge of the scientific community regarding the influenza, indicating which groups of children need timely prevention measures. I consider that this paper would deserve being published. However, there are several major and minor points that should be revised to improve the quality of the manuscript.

Abstract: The sentence “Interstingly, 80.0%.....neurodevelopmental disorders (23.1%)” is a bit confusing. The number of hospitalized children with influenza should be reported here.

The introduction is too short. A short paragraph containing information about influenza should be included.

Table 1. What kind of values are these? For example, in the age (years), the numbers are median and IQR? Please determine. Some of them should be determined in the text as well, again like age (line 83). Abbreviations, like OR, CI or else, should be defined the first time they are used in the test (methods) and in all tables in caption or as a note.

p-value should be written in italics (p-value) everywhere.

In the section of the results (line 83), It is really interesting that Hispanic children were less likely to be hospitalized with influenza than Caucasian children. This comparison is referred in children with comorbidities? How many of them in both race groups have at least one comorbidity and which one was the most common. What was the race in children without any comorbidity?

What about the not-admitted children with lab-confirmed influenza? Is there a statistically significant correlation? I think it should be mentioned.

The influenza vaccination status of children is not analyzed in the study. Do you have this information?

In the discussion, in paragraph with lines 157-164, it is not clear to me if your findings are in agreement with those of the references.

Moreover, the sentence in lines 176-179 is not clear. I do not understand the existence of this extended paragraph for the COVID-19.

Is the information in line 182 in accordance with your findings? Is the cohort of these references a pediatric population? And what race do they have?  

Author Response

See file

Reviewer 4 Report

This paper investigated the impact of comorbidities on the severity of clinical symptoms and the need for hospitalization in children with influenza. This is an important topic and this paper identifies the most important comorbidities affecting hospitalization and length of hospital stay that allow the identification of the most vulnerable patients in the pediatric population. The title is appropriate. The abstract is well written and contains all the relevant information. The introduction is well structured and well clarifies the need to conduct this type of research. The methods are well explained, the statistical tests are properly selected and the results are clearly and correctly presented. The discussion is based on the presented results and supported by relevant data from previous studies. The conclusions are appropriate. In the row 152, "was associated" was 2x repeated. That needs to be corrected

Author Response

See file

Reviewer 5 Report

 Questions to be clarified:

  1. The analysis included patients with laboratory-confirmed cases. However, the probability of being tested could be associated (caused) with the presence of comorbidity: this depends on the capacity of the health care, local guidelines, etc. Is the testing policy influencing the sample of this study?
  2. Is the primary diagnosis for hospitalization influenza? Could it be that the primary reason is, e.g., diabetes control or another problem of a comorbidity?
  3. The length of stay in the hospital depends on health care determinants as well: insurance, coverage, functions, capacity of outpatient services, etc. Are these parameters influencing the results of this study?
  4. Are there any explanations for demographic differences in hospitalization rates (eg male / female, rase difference)?
  5. By stratifying admitted patients by absolute numbers of comorbidities, they are becoming small. Also, the cause-specific confidence intervals of the odds ratios are becoming very wide. Is there statistically significant difference between kind of comorbidity?
  6. For all the results in logistic regression is a univariate analysis used (no adjustment)?
  7. In Table No. 2 what is the reference for age?

Author Response

See file
